# Momentum-Based Policy Gradient with Second-Order Information

**Saber Salehkaleybar**                                                          *s.salehkaleybar@liacs.leidenuniv.nl*
*Leiden Institute of Advanced Computer Science*
*Leiden University*

**Sadegh Khorasani**                                                                      *sadegh.khorasani@epfl.ch*
*School of Computer and Communication Sciences*
*EPFL*

**Negar Kiyavash**                                                                          *negar.kiyavash@epfl.ch*
*College of Management of Technology*
*EPFL*

**Niao He**                                                                                     *niao.he@inf.ethz.ch*
*Department of Computer Science*
*ETH Zurich*

**Patrick Thiran**                                                                          *patrick.thiran@epfl.ch*
*School of Computer and Communication Sciences*
*EPFL*

**Reviewed on OpenReview:** *https://openreview.net/forum?id=2bURaH6RN8*

## Abstract

Variance-reduced gradient estimators for policy gradient methods have been one of the main focus of research in the reinforcement learning in recent years as they allow acceleration of the estimation process. We propose a variance-reduced policy-gradient method, called SHARP, which incorporates second-order information into stochastic gradient descent (SGD) using momentum with a time-varying learning rate. SHARP algorithm is parameter-free, achieving $\epsilon$-approximate first-order stationary point with $O(\epsilon^{-3})$ number of trajectories, while using a batch size of $O(1)$ at each iteration. Unlike most previous work, our proposed algorithm does not require importance sampling, which can compromise the advantage of variance reduction process. Moreover, the variance of estimation error decays at a fast rate of $O(1/t^{2/3})$, where $t$ is the number of iterations. Our extensive experimental evaluations show the effectiveness of the proposed algorithm on various control tasks and its advantage over the state of the art in practice.

## 1 Introduction

Reinforcement Learning (RL) has achieved remarkable success in solving various complex tasks in games (Silver et al., 2017), autonomous driving (Shalev-Shwartz et al., 2016), and robot manipulation (Deisenroth et al., 2013), among other fields. In the RL setting, an agent tries to learn the best actions by interacting with the environment and evaluating its performance based on reward signals. More specifically, in Markov Decision Processes (MDPs), the mathematical formalism for RL, after taking an action, the state changes according to a transition probability model and a reward signal is received based on the action taken and the current state. The main goal of the learner is to find a policy that maps the state space to the action space, maximizing the expected cumulative rewards as the objective function.

Policy gradient methods (Sutton et al., 2000) are often used for training policies in MDPs, especially for high-dimensional continuous action space. In policy gradient methods, the policy is parameterized by an unknown parameter $\theta$ and it is directly optimized using the stochastic first-order gradient of cumulative rewards as it is infeasible to compute the gradient exactly. REINFORCE (Williams, 1992), PGT (Sutton et al., 2000), and GPOMDP (Baxter & Bartlett, 2001) are some classical methods that update the policy by applying a stochastic gradient ascent step. These methods generally require a large number of trajectories due to the large variance of gradient estimates, stemming from randomness of transitions over trajectories.

In the RL literature, several methods have been proposed to reduce the variance in policy gradient methods. For instance, Sutton et al. (2000) proposed to consider a baseline in order to reduce variance of gradient estimation. Konda & Tsitsiklis (2000) presented an actor-critic algorithm that estimates the value function and uses it to mitigate the effect of large variance. Schulman et al. (2016) proposed GAE to control both bias and variance by exploiting a temporal difference relation for the advantage function approximation. More recent work such as TRPO (Schulman et al., 2015) considers a Kullback-Leibler (KL) divergence penalty term in order to ensure that the updated policy remains close to the current policy or PPO (Schulman et al., 2017) that uses clipped surrogate objective function to achieve the same goal. In practice, it has been shown that these algorithms have better performance compared with vanilla policy gradient method.

Most stochastic gradient based policy methods need $O(\epsilon^{-4})$ trajectories in order to achieve $\epsilon$-approximate first-order stationary point ($\epsilon$-FOSP) of the objective function $J(\theta)$, i.e., $\mathbb{E}[\|\nabla J(\theta)\|] \leq \epsilon$ (Ghadimi & Lan, 2013; Shani et al., 2020). In recent years, there have been several attempts to reduce the variance of policy gradient by adapting variance reduction techniques proposed previously in a supervised learning context (a list of previous work is given in Section 4). These methods can achieve sample complexity of $O(\epsilon^{-3})$ in the RL setting and this rate is optimal in stochastic optimization under some mild assumptions on the objective function and stochastic gradients (Arjevani et al., 2020). On the one hand, in supervised learning problems, the objective function is oblivious, in the sense that the randomness that selects the loss function does not depend on the parameters that are to be optimized. On the other hand, in the RL setting, the distribution over trajectories is non-stationary and changes over time as the parameters of policy are updated. To resolve this issue, most previous work use importance sampling techniques, which may degrade the effectiveness of the variance reduction process (Yang et al., 2022). Moreover, to analyze the convergence rate of these methods, a strong assumption on the variance of importance sampling weights is made, which may not hold in the RL setting. Most importantly, these methods often need huge batch sizes, which is highly undesirable in practice.

In this paper, we propose the Stochastic Hessian Aided Recursive Policy gradient (SHARP) algorithm, which incorporates second-order information into SGD with momentum. Our main contributions are summarized as follows:

- Under some common regularity assumptions on the parameterized policy, SHARP reaches $\epsilon$-FOSP with a sample complexity of $O(\epsilon^{-3})$. Moreover, our algorithm does not use importance sampling techniques. As a result, we can relax the strong additional assumptions on importance sampling weights, which are customary in the literature.

- The batch size of SHARP is $O(1)$ and it does not require checkpoints, thanks to the use of a second-order term in the updates and time-varying learning rate and momentum weight.

- SHARP is parameter-free in the sense that the initial learning rate and momentum weight do not depend on the parameters of the problem. Moreover, the variance of the estimation error decays with a rate of $O(1/t^{2/3})$, where $t$ is the number of iterations.

- Our experimental results show that SHARP outperforms the state of the art on various control tasks, with remarkable performance in more complex environments.

The rest of this paper is organized as follows: In Section 2, we define the problem and provide some notations and background on variance reduction methods in supervised learning. In Section 3, we describe the proposed algorithm, SHARP, and we analyze its convergence rate. In Section 4, we give a summary of previous work

and discuss how SHARP differs from them. In Section 5, we evaluate the performance of SHARP against the related work experimentally. Finally, we conclude the paper in Section 6.

## 2 Preliminaries

### 2.1 Notations and problem definition

Consider a discrete-time MDP $\mathcal{M} = \{\mathcal{S}, \mathcal{A}, P, R, \gamma, \rho\}$ that models how an agent interacts with a given environment. $\mathcal{S}$ and $\mathcal{A}$ are state space and action space, respectively. $P(s'|s, a)$ denotes the probability of transiting to state $s'$ from $s$ after taking action $a$. The reward function $R$ returns reward $r(s, a)$ when action $a$ is taken in state $s$. Parameter $\gamma \in (0, 1)$ denotes the discount factor and $\rho$ is the distribution of starting state. The actions are chosen according to policy $\pi$ where $\pi(a|s)$ is the probability of taking action $a$ for a given state $s$. Here, we assume that the policy is parameterized with a vector $\theta \in \mathbb{R}^d$ and use shorthand notation $\pi_\theta$ for $\pi_\theta(a|s)$.

For a given time horizon $H$, according to policy $\pi_\theta$, the agent observes a sequence of state-action pairs $\tau = (s_0, a_0, \cdots, s_{H-1}, a_{H-1})$ called a trajectory. The probability of observing a trajectory $\tau$ for a given policy $\pi_\theta$ is:

$$p(\tau|\pi_\theta) = \rho(s_0) \prod_{h=0}^{H-1} P(s_{h+1}|s_h, a_h)\pi_\theta(a_h|s_h). \tag{1}$$

The discounted cumulative reward for a trajectory $\tau$ is defined as $R(\tau) := \sum_{h=0}^{H-1} \gamma^h r(s_h, a_h)$ and the expected return for a policy $\pi_\theta$ is:

$$J(\theta) := \mathbb{E}_{\tau \sim \pi_\theta}[R(\tau)]. \tag{2}$$

The main goal in policy-based RL is to find $\theta^* = \arg\max_\theta J(\theta)$. As in many applications, $J(\theta)$ is non-convex and we settle instead for obtaining $\epsilon$-FOSP $\hat{\theta}$ such that $\mathbb{E}[\|\nabla J(\hat{\theta})\|] \leq \epsilon$. It can be shown that:

$$\nabla J(\theta) = \mathbb{E}\left[\sum_{h=0}^{H-1} \Psi_h(\tau)\nabla \log \pi_\theta(a_h|s_h)\right], \tag{3}$$

where $\Psi_h(\tau) = \sum_{t=h}^{H-1} \gamma^t r(s_t, a_t)$. Therefore, for any trajectory $\tau$, $g(\tau; \theta) := \sum_{h=0}^{H-1} \Psi_h(\tau)\nabla \log \pi_\theta(a_h|s_h)$ is an unbiased estimator of $\nabla J(\theta)$. The vanilla policy gradient updates $\theta$ as follows:

$$\theta \leftarrow \theta + \eta g(\tau; \theta), \tag{4}$$

where $\eta$ is the learning rate.

The Hessian matrix of $J(\theta)$ can be written as follows (Shen et al., 2019):

$$\nabla^2 J(\theta) = \mathbb{E}[\nabla\Phi(\theta; \tau)\nabla \log p(\tau|\pi_\theta)^T + \nabla^2\Phi(\theta; \tau)], \tag{5}$$

where $\Phi(\theta; \tau) = \sum_{h=0}^{H-1}\sum_{t=h}^{H-1} \gamma^t r(s_t, a_t) \log \pi_\theta(a_h|s_h)$. For a given trajectory $\tau$, $B(\tau; \theta) := \nabla\Phi(\theta; \tau)\nabla \log p(\tau|\pi_\theta)^T + \nabla^2\Phi(\theta; \tau)$ is an unbiased estimator of the Hessian matrix.

### 2.2 Variance reduced methods for gradient estimation

Variance reduced methods for estimating the gradient vector were originally proposed for the stochastic optimization setting:

$$\min_{\theta \in \mathbb{R}^d} \mathbb{E}_{z \sim p(z)}[f(\theta, z)], \tag{6}$$

where a sample $z$ is drawn from distribution $p(z)$ and $f(\cdot, z)$ is commonly assumed to be smooth and non-convex function of $\theta$. This setting is mainly considered in supervised learning contexts where $\theta$ corresponds to the parameters of the training model and $z = (x, y)$ is the training sample, with $x$ the feature vector of the sample and $y$ the corresponding label. In this setting, the distribution $p(z)$ is invariant with respect to parameter $\theta$.

---

**Algorithm 1** Common framework in variance reduction methods

---
1: **for** $t = 0, \cdots, T - 1$ **do**
2:

$$
h_t = \begin{cases} \dfrac{1}{|\mathcal{B}_{check}|} \displaystyle\sum_{z \in \mathcal{B}_{check}} \nabla f(\theta_t, z) & \text{if } t \equiv 0 \pmod{Q}, \qquad\qquad (7) \\[2em] h_{t-1} + \dfrac{1}{|\mathcal{B}|} \displaystyle\sum_{z \in \mathcal{B}} \left( \nabla f(\theta_t, z) - \nabla f(\theta_{t-1}, z) \right), & \text{otherwise.} \qquad\qquad (8) \end{cases}
$$

3: $\quad \theta_{t+1} \leftarrow \theta_t - \eta h_t$
4: **end for**
5: Return $\theta_t$ with $t$ chosen randomly from $\{0, \cdots, T - 1\}$

---

The common approach for reducing the variance of gradient estimation is to reuse past gradient vectors. The pseudo-code for this general framework for variance reduction is given in Algorithm 1. After every pre-determined number of iterations $Q$, there is a checkpoint to obtain an unbiased estimate of the gradient, denoted by $h_t$, at the current parameter $\theta_t$ by taking a batch of samples $\mathcal{B}_{check}$. Between any two consecutive checkpoints, the gradient at the parameter $\theta_t$ is estimated according to equation 8 by taking a batch of samples $\mathcal{B}$ drawn from $p(z)$. The above framework appeared in several previous variance reduction methods in stochastic optimization such as SARAH (Nguyen et al., 2017) and SPIDER (Fang et al., 2018). Zhang (2021) discusses how to choose the size of batches and the parameters $Q$ and $\eta$. In fact, there is a trade-off between $\eta$ and $|\mathcal{B}|$. If a small batch size is used, then $\eta$ is also required to be small. The two extremes are SpiderBoost (Wang et al., 2019) ($|\mathcal{B}| = O(\epsilon^{-1}), \eta = O(1)$) and SARAH (Nguyen et al., 2017) ($|\mathcal{B}| = O(1)$, $\eta = O(\epsilon)$). Very recently, Li et al. (2021) proposed PAGE, where in each iteration $t$, either a batch of samples is taken with probability $p_t$ to update the gradient or the previous estimate of the gradient is used with a small adjustment, with probability $1 - p_t$.

In the context of RL, a sample $z$ corresponds to a trajectory $\tau$. Unlike supervised learning, the distribution of these trajectories depends on the parameters of the policy that generates them. Therefore, in the second term in the sum in equation 8, namely $\nabla f(\theta_{t-1}, z)$, $z$ (i.e., the trajectory $\tau$ in the RL context) is generated according to policy $\pi_{\theta_t}$ while $\theta_{t-1}$ is the parameter of the policy at the previous iteration. In the RL setting, importance sampling technique is commonly used to account for the distribution shift as follows:

$$
h_t = h_{t-1} + \frac{1}{|\mathcal{B}|} \sum_{\tau \in \mathcal{B}} g(\theta_t; \tau) - w(\tau | \theta_t, \theta_{t-1}) g(\theta_{t-1}; \tau), \tag{9}
$$

with the weights $w(\tau | \theta_t, \theta_{t-1}) = \prod_{h=0}^{H-1} \frac{\pi_{\theta_{t-1}}(a_h | s_h)}{\pi_{\theta_t}(a_h | s_h)}$.

As we shall see in Section 4, nearly all variance reduction approaches in RL employing the general framework of Algorithm 1 use an importance sampling technique. This could significantly degrade the performance of the approach as the gradient estimates depend heavily on these weights (Yang et al., 2022). Besides, these variance reduction methods often need giant batch sizes at checkpoints, which is not practical in the RL setting. Finally, the hyper-parameters of these approaches must be selected carefully as they often use non-adaptive learning rates. To resolve the issue of requiring huge batch-sizes, in the context of stochastic optimization, a variance reduction method called STORM (Cutkosky & Orabona, 2019) was proposed with the following update rule:

$$
h_t = (1 - \alpha_t)h_{t-1} + \alpha_t \nabla f(\theta_t, z_t) + (1 - \alpha_t)(\nabla f(\theta_t, z_t) - \nabla f(\theta_{t-1}, z_t))
$$
$$
\theta_{t+1} \leftarrow \theta_t - \eta_t h_t, \tag{10}
$$

where $z_t$ is the sample drawn at iteration $t$ and $\alpha_t$ and $\eta_t$ are the adaptive momentum weight and learning rate, respectively. Compared with SGD with momentum, the main difference in STORM is the correction term $\nabla f(\theta_t, z_t) - \nabla f(\theta_{t-1}, z_t)$ in equation 10. Cutkosky & Orabona (2019) showed that by adaptively updating $\alpha_t$ and $\eta_t$ based on the norm of stochastic gradient in previous iterations, STORM can achieve

---

**Algorithm 2** The SHARP algorithm

---

**Input:** Initial point $\theta_0$, parameters $\alpha_0, \eta_0$, and number of iterations $T$

1: Sample trajectory $\tau_0$ with policy $\pi_{\theta_0}$
2: $v_0 \leftarrow g(\tau_0; \theta_0)$
3: $\theta_1 \leftarrow \theta_0 + \eta_0 \frac{v_0}{\|v_0\|}$
4: **for** $t = 1, \cdots, T - 1$ **do**
5:     Sample $b_t \sim U(0, 1)$
6:     $\theta_t^b \leftarrow b_t \theta_t + (1 - b_t) \theta_{t-1}$
7:     Sample trajectories $\tau_t$ and $\tau_t^b$ with policies $\pi_{\theta_t}$ and $\pi_{\theta_t^b}$, respectively
8:     $\eta_t \leftarrow \frac{\eta_0}{t^{2/3}}$, $\alpha_t \leftarrow \frac{\alpha_0}{t^{2/3}}$
9:     $v_t \leftarrow (1 - \alpha_t)(v_{t-1} + B(\tau_t^b; \theta_t^b)(\theta_t - \theta_{t-1})) + \alpha_t g(\tau_t; \theta_t)$
10:    $\theta_{t+1} \leftarrow \theta_t + \eta_t \frac{v_t}{\|v_t\|}$
11: **end for**
12: Return $\theta_t$ with $t$ chosen randomly from $\{0, \cdots, T - 1\}$

---

the same convergence rate as previous methods without requiring checkpoints nor a huge batch size. Later, a parameter-free version, called STORM+ (Levy et al., 2021), has been introduced using new adaptive learning rate and momentum weight. However, to adapt these methods in the RL setting, we still need to use importance sampling techniques because of the term $\nabla f(\theta_{t-1}, z_t)$. Recently, Tran & Cutkosky (2022) showed that the correction term can be replaced with a second-order term $\nabla^2 f(\theta_t, z_t)(\theta_t - \theta_{t-1})$ by making the additional assumption that the objective function is second-order smooth. Besides, the above Hessian vector product can be computed in $O(Hd)$ (similar to the computational complexity of obtaining the gradient vector) by executing Pearlmutter's algorithm (Pearlmutter, 1994).

## 3 The SHARP Algorithm

In this section, we propose the SHARP algorithm, which incorporates second-order information into SGD with momentum and we provide a convergence guarantee. SHARP is described in Algorithm 2. At each iteration $t$, we draw sample $b_t$ from a uniform distribution in the interval $[0, 1]$ (line 5) and next obtain $\theta_t^b$ as the linear combination of $\theta_{t-1}$ and $\theta_t$ with coefficients $1 - b_t$ and $b_t$ (line 6). In line 7, we sample trajectories $\tau_t$ and $\tau_t^b$ according to policies $\pi_{\theta_t}$ and $\pi_{\theta_t^b}$, respectively. Afterwards, we update the momentum weight $\alpha_t$ and the learning rate $\eta_t$ (line 8) and then compute the estimate of gradient at time $t$, i.e., $v_t$, using the Hessian vector product $B(\tau_t^b; \theta_t^b)(\theta_t - \theta_{t-1})$ (please refer to Section 2.1 for the definition of $B(\tau; \theta)$) and stochastic gradient $g(\tau_t; \theta_t)$ (line 9). Finally we update $\theta_t$ based on a normalized version of $v_t$ in line 10.

Note that in each iteration, we compute one Hessian vector product, which can be done with the same computational complexity of obtaining the gradient vector using Pearlmutter's algorithm (Pearlmutter, 1994). Therefore, the computational cost of SHARP per iteration is in the same order as the one for first-order methods such as REINFORCE (Williams, 1992).

**Remark 1** *By choosing a point uniformly at random on the line between $\theta_{t-1}$ and $\theta_t$, we can ensure that $B(\tau_t^b; \theta_t^b)(\theta_t - \theta_{t-1}))$ is an unbiased estimate of $\nabla J(\theta_t) - \nabla J(\theta_{t-1})$ (see equation 22 in Appendix A). As we mentioned before, in the context of stochastic optimization, Tran & Cutkosky (2022) used the second-order term $\nabla^2 f(\theta_t, z_t)(\theta_t - \theta_{t-1})$, which is biased as the Hessian vector product is evaluated at the point $\theta_t$. As a result, in order to provide the convergence guarantee, it is further assumed that the objective function is second-order smooth in (Tran & Cutkosky, 2022).*

**Remark 2** *To give an intuition why the second-order term is helpful in the update in line 9, consider the following error term:*

$$\epsilon_t = v_t - \nabla J(\theta_t). \tag{11}$$

We can rewrite the above error term as follows:

$$
\begin{aligned}
\epsilon_t = (1 - \alpha_t)(v_{t-1} - \nabla J(\theta_t) + B(\tau_t^b; \theta_t^b)(\theta_t - \theta_{t-1})) \\
+ \alpha_t(g(\tau_t; \theta_t) - \nabla J(\theta_t)).
\end{aligned}
\tag{12}
$$

Now, for a moment, suppose that $\mathbb{E}[v_{t-1}] = \mathbb{E}[\nabla J(\theta_{t-1})]$ (with total expectation on both sides). Then,

$$
\mathbb{E}[v_{t-1} - \nabla J(\theta_t) + B(\tau_t^b; \theta_t^b)(\theta_t - \theta_{t-1})] = 0.
\tag{13}
$$

As $v_0$ is an unbiased estimate of gradient at $\theta_0$, we can easily show by induction that according to above equation, $\mathbb{E}[v_t] = \mathbb{E}[\nabla J(\theta_t)]$ for any $t \geq 0$.

In the next part, we provide a theoretical guarantee on the convergence rate of SHARP algorithm.

## 3.1 Convergence Analysis

In this part, we analyze the convergence rate of Algorithm 2 under bounded reward function and some regularity assumptions on the policy $\pi_\theta$.

**Assumption 1 (Bounded reward)** *For $\forall s \in \mathcal{S}, \forall a \in \mathcal{A}, |R(s, a)| < R_0$ where $R_0 > 0$ is some constant.*

**Assumption 2 (Parameterization regularity)** *There exist constants $G, L > 0$ such that for any $\theta \in \mathbb{R}^d$ and for any $s \in \mathcal{S}, a \in \mathcal{A}$:*
*(a) $\|\nabla \log \pi_\theta(a|s)\| \leq G$,*
*(b) $\|\nabla^2 \log \pi_\theta(a|s)\| \leq L$.*

Assumptions 1 and 2 are common in the RL literature (Papini et al., 2018; Shen et al., 2019) to analyze the convergence of policy gradient methods. Under these assumptions, the following upper bounds can be derived on $\mathbb{E}[\|g(\tau; \theta) - \nabla J(\theta)\|^2]$ and $\mathbb{E}[\|B(\tau; \theta) - \nabla^2 J(\theta)\|^2]$.

**Lemma 1 (Shen et al. (2019))** *Under Assumptions 1 and 2:*

$$
\begin{aligned}
\mathbb{E}[\|g(\tau; \theta) - \nabla J(\theta)\|^2] \leq \sigma_g^2 \\
\mathbb{E}[\|B(\tau; \theta) - \nabla^2 J(\theta)\|^2] \leq \sigma_B^2,
\end{aligned}
\tag{14}
$$

*where $\sigma_g^2 = \frac{G^2 R_0^2}{(1-\gamma)^4}$ and $\sigma_B^2 = \frac{H^2 G^4 R_0^2 + L^2 R_0^2}{(1-\gamma)^4}$.*

Based on these bounds, we can provide the following guarantee on the convergence rate of SHARP algorithm. All proofs are provided in the appendix.

**Theorem 1** *Suppose that the initial momentum weight $\alpha_0 \in (2/3, 1]$ and initial learning rate $\eta_0 > 0$. Under Assumptions 1 and 2, Algorithm 2 guarantees that:*

$$
\mathbb{E}\left[\frac{1}{T} \sum_{t=1}^T \|\nabla J(\theta_t)\|\right] \leq \frac{8\sqrt{C} + 9C_J/\eta_0}{T^{1/3}} + \frac{6\sigma_B \eta_0}{T^{2/3}},
\tag{15}
$$

*where $C = 3\alpha_0(48\sigma_B^2 \eta_0^2/\alpha_0 + (6\alpha_0 + 1/\alpha_0)\sigma_g^2)/(3\alpha_0 - 2)$ and $C_J = R_0/(1 - \gamma)$.*

**Corollary 1** *The right hand side of equation 15 is dominated by the first term. If we set $\eta_0$ in the order of $\sqrt{C_J/\sigma_B}$, then the number of trajectories for achieving $\epsilon$-FOSP would be $O(\frac{1}{(1-\gamma)^2 \epsilon^3})$, where we assume that the time horizon $H$ is set in the order of $1/(1 - \gamma)$.*

**Remark 3** *Let us rewrite equation 12 as follows:*

$$
\epsilon_t = (1 - \alpha_t)\epsilon_{t-1} + \alpha_t U_t + (1 - \alpha_t)W_t,
$$

*where $U_t = (g(\tau_t; \theta_t) - \nabla J(\theta_t))$ and $W_t = B(\tau_t^b; \theta_t^b)(\theta_t - \theta_{t-1}) - (\nabla J(\theta_t) - \nabla J(\theta_{t-1}))$. As $g(\tau_t; \theta_t)$ and $B(\tau_t^b; \theta_t^b)(\theta_t - \theta_{t-1})$ are unbiased estimates of $\nabla J(\theta_t)$ and $\nabla J(\theta_t) - \nabla J(\theta_{t-1})$, respectively, we have $\mathbb{E}[U_t] = 0$ and $\mathbb{E}[W_t] = 0$, which in turn implies that $\mathbb{E}[\langle \epsilon_{t-1}, U_t \rangle] = \mathbb{E}[\langle \epsilon_{t-1}, W_t \rangle] = 0$. After some simple manipulations (see Appendix A for more details), we have*

$$\mathbb{E}[\|\epsilon_t\|^2] \leq (1 - \alpha_t)\mathbb{E}[\|\epsilon_{t-1}\|^2] + 2\alpha_t^2 \mathbb{E}[\|U_t\|^2] + 2\mathbb{E}[\|W_t\|^2].$$

*Now, according to Lemma 1, we know that $\mathbb{E}[\|U_t\|^2] \leq \sigma_g^2$. Moreover, it can be shown that $\|B(\tau; \theta)\| \leq \sigma_B$ for any trajectory $\tau$ (Shen et al., 2019). Therefore, $\|W_t\| \leq 2\sigma_B \|\theta_t - \theta_{t-1}\|$. Moreover, the normalized update of SHARP yields that $\|\theta_t - \theta_{t-1}\|^2 = \eta_{t-1}^2$. Hence,*

$$\mathbb{E}[\|\epsilon_t\|^2] \leq (1 - \alpha_t)\mathbb{E}[\|\epsilon_{t-1}\|^2] + 2\alpha_t^2 \sigma_g^2 + 8\sigma_B^2 \eta_{t-1}^2.$$

*As $\alpha_t$ and $\eta_t$ have the same dependency on $t$, along the iterations of the SHARP algorithm, the following inequality holds for any $t \geq 1$:*

$$\mathbb{E}[\|\epsilon_t\|^2] \leq (1 - \alpha_t)\mathbb{E}[\|\epsilon_{t-1}\|^2] + O(\eta_t^2). \tag{16}$$

*Therefore, the variance of the estimation error decays with rate $O(1/t^{2/3})$ (see Appendix B for the proof). To the best of our knowledge, existing variance reduction methods only guarantee the decay of cumulative variance. This appealing property of SHARP is largely due to the use of unbiased Hessian-aided gradient estimator and normalized gradient descent. Moreover, as a byproduct of these desirable properties, our convergence analysis turns out to be more simple, compared to existing work (Cutkosky & Orabona, 2019; Tran & Cutkosky, 2022). This could be of independent interest for better theory of variance-reduced methods.*

**Remark 4** *The SHARP algorithm is parameter-free in the sense that $\alpha_0$ and $\eta_0$ are constants that do not depend on other parameters in the problem. Therefore, for any choice of $2/3 < \alpha_0 \leq 1$ and $\eta_0 > 0$, we can guarantee convergence to $\epsilon$-FOSP with the sample complexity of $O(\epsilon^{-3})$. However, in practice, it is desirable to tune these constants to have smaller constants in the numerators of convergence rates in equation 15. For instance, $\sigma_B$ might be large in some RL settings and we control the constant in the first term on the right hand side of equation 15 by tuning $\eta_0$. It is noteworthy that STORM+ is also parameter-free, but it requires adaptive learning rate and momentum weight that depend on stochastic gradients in previous iterations.*

**Remark 5** *Regarding the dependency on $\epsilon$, in the context of stochastic optimization, Arjevani et al. (2020) have shown that under some mild assumptions on the objective function and stochastic gradient, the rate of $O(1/\epsilon^3)$ is optimal in order to obtain $\epsilon$-FOSP, and cannot be improved with stochastic p-th order methods for $p \geq 2$.*

## 4  Related Work

In recent years, several variance-reduced methods have been proposed in order to accelerate the existing PG methods. Papini et al. (2018) and Xu et al. (2020) proposed SVRPG algorithm based on SVRG (Johnson & Zhang, 2013), with sample complexity of $O(1/\epsilon^4)$ and $O(1/\epsilon^{10/3})$, respectively. This algorithm requires importance sampling techniques as well as the following additional assumption for guaranteeing convergence to $\epsilon$-FOSP:

- Bounded variance of importance sampling weights: For any trajectory $\tau$, it is assumed that:

$$Var\left(\frac{p(\tau | \pi_{\theta_1})}{p(\tau | \pi_{\theta_2})}\right) \leq W, \qquad \forall \theta_1, \theta_2 \in \mathbb{R}^d, \tag{17}$$

   where $W < \infty$ is a constant.

The above assumption is fairly strong as the importance sampling weight could grow exponentially with horizon length $H$ (Zhang et al., 2021). In order to remove importance sampling weights, Shen et al. (2019)

| Method | SC | $|\mathcal{B}|$ | $|\mathcal{B}_{check}|$ | Checkpoint | IS | Further Assump. |
|---|---|---|---|---|---|---|
| SVRPG (Xu et al., 2020) | $O(\frac{1}{\epsilon^{10/3}})$ | $O(\frac{1}{\epsilon^{4/3}})$ | $O(\frac{1}{\epsilon^{4/3}})$ | Needed | Needed | Assump. in equation 17 |
| HAPG (Shen et al., 2019) | $O(\frac{1}{\epsilon^3})$ | $O(\frac{1}{\epsilon})$ | $O(\frac{1}{\epsilon^2})$ | Needed | Not needed | - |
| SRVR-PG (Xu et al., 2019) | $O(\frac{1}{\epsilon^3})$ | $O(\frac{1}{\sqrt{\epsilon}})$ | $O(\frac{1}{\epsilon})$ | Needed | Needed | Assump. in equation 17 |
| HSPGA (Pham et al., 2020) | $O(\frac{1}{\epsilon^3})$ | $O(1)$ | - | Not needed | Needed | Assump. in equation 17 |
| MBPG (Huang et al., 2020) | $\tilde{O}(\frac{1}{\epsilon^3})$ | $O(1)$ | - | Not needed | Needed | Assump. in equation 17 |
| VRMPO (Yang et al., 2022) | $O(\frac{1}{\epsilon^3})$ | $O(\frac{1}{\epsilon})$ | $O(\frac{1}{\epsilon^2})$ | Needed | Not needed | - |
| VR-BGPO (Huang et al., 2022) | $O(\frac{1}{\epsilon^3})$ | $O(1)$ | - | Not Needed | Needed | Assump. in equation 17 |
| PAGE-PG (Gargiani et al., 2022) | $O(\frac{1}{\epsilon^3})$ | $O(1)$ | $O(\frac{1}{\epsilon^2})$ | Needed | Needed | Assump. in equation 17 |
| This paper | $O(\frac{1}{\epsilon^3})$ | $O(1)$ | - | Not needed | Not needed | - |

Table 1: Comparison of main variance-reduced policy gradient methods to achieve $\epsilon$-FOSP based on sample complexity (SC), batch size ($|\mathcal{B}|$), batch size at checkpoints ($|\mathcal{B}_{check}|$), and the need for checkpoints, importance sampling (IS), and additional assumptions.

proposed the HAPG algorithm, which uses second-order information and achieves better sample complexity of $O(1/\epsilon^3)$. However, it still needs checkpoints and large batch sizes of $|\mathcal{B}| = O(1/\epsilon)$, and $|\mathcal{B}_{check}| = O(1/\epsilon^2)$.

In Table 1, we compare the main variance-reduced policy gradient methods achieving $\epsilon$-FOSP in terms of sample complexity and batch size[1]. In this table, after HAPG (Shen et al., 2019), all the proposed variance reduction methods achieve a similar sample complexity. The orders of batch sizes are also the same as in HAPG. Xu et al. (2019) proposed SRVR-PG, and used stochastic path-integrated differential estimators for variance reduction. This algorithm uses important sampling weights and the required batch sizes are in the order of $|\mathcal{B}| = O(1/\sqrt{\epsilon})$ and $|\mathcal{B}_{check}| = O(1/\epsilon)$. Later, Pham et al. (2020) proposed HSPGA by adapting the SARAH estimator for reducing the variance of REINFORCE. HSPGA still needs importance sampling weights, but the batch size is reduced to $O(1)$. Huang et al. (2020) proposed three variants of momentum-based policy gradient (called MBPG), which are based on the STORM algorithm (Cutkosky & Orabona, 2019). Thus, the required batch size is $O(1)$, similarly to STORM. However, it still needs to use importance sampling weights. (Zhang et al., 2020) used a similar update for the estimate of stochastic gradient as the one in SHARP for Frank-Wolfe type algorithms in the context of constrained optimization. Later, Zhang et al. (2021) proposed TSIVR-PG with a gradient truncation mechanism in order to resolve some issues pertaining to the use of importance sampling weights. In their convergence analysis, they are restricted to soft-max policy with some specific assumptions on the parameterization functions. More recently, two methods using mirror-descent algorithm based on Bregman divergence, called VR-BGPO (Huang et al., 2022) and VR-MPO (Yang et al., 2022) have been proposed, achieving $\epsilon$-FOSP if the mirror map in Bregman divergence is the $l_2$-norm. Very recently, based on PAGE (Li et al., 2021), Gargiani et al. (2022) proposed the PAGE-PG algorithm which takes a batch of $O(\epsilon^{-2})$ samples for updating the parameters with probability $p_t$ or reuse the previous estimate gradient with a small adjustment, with probability $1 - p_t$. The proposed algorithm requires importance sampling weights and thus the additional assumption in equation 17 to guarantee convergence to $\epsilon$-FOSP with a sample complexity of $O(\epsilon^{-3})$.

---

[1]Please note that the sample complexity also depends on the other parameters such as horizon length $H$ and discount factor $\gamma$. Here, we just mention the dependency of sample complexity on $\epsilon$.

There are few recent work on the global convergence of policy gradient methods. For instance, Liu et al. (2020) showed global convergence of policy gradient, natural policy gradient, and their variance reduced variants, in the case of positive definite Fisher information matrix of the policy. Chung et al. (2021) studied the impact of baselines on the learning dynamics of policy gradient methods and showed that using a baseline minimizing the variance can converge to a sub-optimal policy. Recently, Ding et al. (2022) studied the soft-max and the Fisher non-degenerate policies, and showed that adding a momentum term improves the global optimality sample complexities of vanilla PG methods by $\tilde{O}(\epsilon^{-1.5})$ and $\tilde{O}(\epsilon^{-1})$, respectively.

The main methods discussed above are summarized in Table 1. For each method, we mention whether it needs checkpoints and/or importance sampling weights.[2] All of them require Assumptions 1 and 2. In the last column, additional assumptions besides these two are listed for each method.

Comparing the sample complexity of the SHARP algorithm with previous work, note that all the algorithms (including SHARP) under SVRPG in Table 1 achieve rates of $O(1/\epsilon^3)$ or $\tilde{O}(1/\epsilon^3)$. Without any further assumption, SHARP is the only method that requires no checkpoints, no importance sampling weights, and has a batch size of the order of $O(1)$. As we will see in the next section, besides these algorithmic advantages, it has remarkable performance compared to the state of the art in various control tasks.

## 5 Experiments

In this section, we evaluate the performance of the proposed algorithm and compare it with previous work for control tasks in MuJoCo simulator (Todorov et al., 2012) which is a physical engine, suitable for simulating robotic tasks with good accuracy and speed in the RL setting. We implemented SHARP in the Garage library (garage contributors, 2019) as it allows for maintaining and integrating it in future versions of Garage library for easier dissemination. We utilized a Linux server with Intel Xeon CPU E5-2680 v3 (24 cores) operating at 2.50GHz with 377 GB DDR4 of memory, and Nvidia Titan X Pascal GPU. The implementation of SHARP is available as supplementary material.

We considered the following four control tasks with continuous action space: Reacher, Walker, Humanoid, and Swimmer. In Reacher, an arm with two degrees of freedom aims at reaching a target point in a two-dimensional plane. A higher reward is attained if the arm gets closer to the target point. In Walker, a humanoid walker tries to move forward in a two dimensional space, i.e., it can only fall forward or backward. The state contains velocities of different parts of body and joint angles and the actions represent how to move legs and foot joints. The reward signal is based on the current velocity of the agent. In Humanoid, a three-dimensional bipedal robot is trained to walk forward as fast as possible, without falling over. The state space is 376-dimensional, containing the position and velocity of each joint, the friction of the actuator, and contact forces. The action space is a 17-dimensional continuous space. Finally, in Swimmer, the agent is in a two-dimensional pool and the goal is to move as fast as possible in the right direction.

We compared the SHARP algorithm with PG methods that provide theoretical guarantees for converging to an approximate FOSP: PAGE-PG (Gargiani et al., 2022), IS-MBPG (Huang et al., 2020) which is based on STORM, HAPG (Shen et al., 2019) which does not require IS weights, and VR-BGPO (Huang et al., 2022) which is a mirror descent based algorithm. We also considered REINFORCE (Sutton et al., 2000) as a baseline algorithm. There are some other approaches in the literature with theoretical guarantees such as VRMPO (Yang et al., 2022), and STORM-PG (Yuan et al., 2020) but the official implementations are not publicly available and our request to access the code from the authors remained unanswered.

For each algorithm, we used the same set of Gaussian policies parameterized with neural networks having two layers of 64 neurons each. Baselines and environment settings (such as maximum trajectory horizon, and reward intervals) were considered the same for all algorithms. We chose a maximum horizon of 500 for Walker, Swimmer, and Humanoid and 50 for Reacher. More details about experiments are provided in Appendix E.

---

[2]To be more precise, although PAGE-PG, has no fixed checkpoints, it takes a batch of $O(\epsilon^{-2})$ to get an unbiased estimate of the gradient with probability $p_t$. Therefore, in this sense, it requires checkpoints.

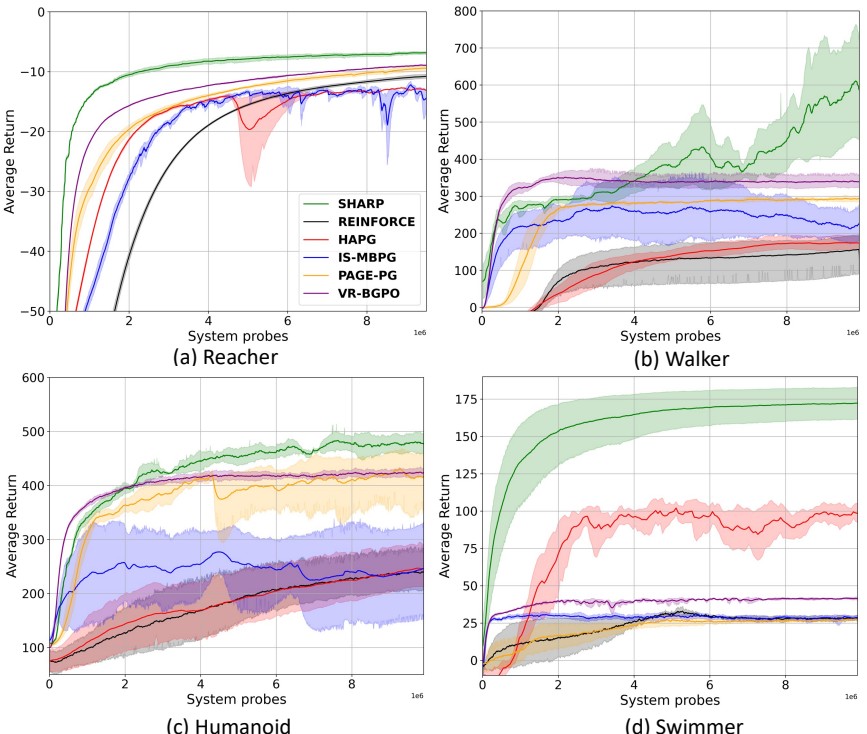

Figure 1: Comparison of SHARP with other variance reduction methods on four control tasks.

In the literature, it has been observed that most PG methods are quite sensitive to parameter initialization or random seeds (Henderson et al., 2018). Hence, it might be challenging in some cases to reproduce previous results. Moreover, it is not clear how to compare methods in terms of performance (e.g., the average episode return) and robustness (such as the standard deviation (STD) of return) at the same time.

To resolve the above issues, we considered the performance-robustness (PR) metric proposed in (Khorasani et al., 2023), capturing both the average return and STD of return of an algorithm. In particular, for any algorithm $A$, after observing $t$ state-actions pairs (which we call system probes), the lower bound of the confidence interval 90 percent of average return over $n$ runs of the algorithm (denoted by $LCI_A(n, t)$) is computed.[3] The PR metric is defined by averaging $LCI_A(n, t)$ over all system probes $t = 1, \cdots, T$ as follows:

$$PR_A(n) = \frac{1}{T} \sum_{t=1}^{T} LCI_A(n, t), \tag{18}$$

where $T$ is maximum number of system probes.

We used grid search to tune the hyper-parameters of all the considered algorithms. For the algorithms (except SHARP), the search space for each hyper-parameter was chosen based on the one from the original papers. For each configuration of the hyper-parameters, we ran each algorithm $A$, five times and computed $PR_A(5)$. We selected the configuration which maximized $PR_A(5)$ and then reported $PR_A(10)$ of each algorithm for the selected configuration based on 10 different runs in Table 2. SHARP achieved the highest PR in all environments compared with the other algorithms. In Appendix D, we also reported the PR metric by considering the upper bounds of confidence intervals instead of lower bounds (i.e., the "best-case" performance

---

[3]As recommended in (Agarwal et al., 2021), for $n = 10$ available runs, we used percentile bootstrap to compute 90% confidence interval with 2000 number of bootstrap samples.

Table 2: Comparison of SHARP with other variance-reduced methods in terms of PR. In each environment, the highest PR is in bold.

|  | Reacher | Walker | Humanoid | Swimmer |
|---|---|---|---|---|
| HAPG | -25.30 | 94.95 | 137.59 | 68.22 |
| IS-MBPG | -22.45 | 175.58 | 164.55 | 26.75 |
| PAGE-PG | -18.67 | 243.63 | 335.24 | 17.65 |
| REINFORCE | -30.31 | 51.03 | 140.06 | 17.84 |
| VR-BGPO | -16.15 | 313.42 | 393.90 | 37.92 |
| SHARP (our algorithm) | **-10.97** | **323.63** | **415.92** | **141.42** |

rather than the "worst-case" performance). Still, SHARP outperforms other algorithms in all environments. This is also evident from Figure 1 where the confidence interval of SHARP is fairly tight.

We considered the confidence interval of the performance (average return) to show the sensitivity of an algorithm to random seeds. As can be seen in Figure 1, SHARP not only achieves the highest average return after 10 million system probes but also has a relatively small confidence interval.

## 6 Conclusion

We proposed a variance-reduced policy-gradient method, which incorporates second-order information, i.e., Hessian vector product, into SGD with momentum. The Hessian vector product can be computed with an efficiency that is similar to that of obtaining the gradient vector. Therefore, the computational complexity per iteration of the proposed algorithm, SHARP, remains in the same order as first-order methods. More importantly, using the second-order correction term enables us to obtain an estimate of the gradient and to completely bypass importance sampling. Moreover, the batch size is $O(1)$ and there is no need for checkpoints, which makes the algorithm appealing in practice. Under some regularity assumptions on the parameterized policy, we showed that it achieves $\epsilon$-FOSP with sample complexity of $O(\epsilon^{-3})$. SHARP is parameter-free in the sense that the initial learning rate and momentum weight do not depend on the parameters of the problem. Experimental results show its remarkable performance in various control tasks, especially in some complex environments, such as Humanoid, compared to the state of the art.

### Acknowledgments

This research was in part supported by the Swiss National Science Foundation under NCCR Automation, grant agreement 51NF40_180545 and Swiss SNF project 200021_204355/1.

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

## A    Proof of Theorem 1

Let us define $\epsilon_t := v_t - \nabla J(\theta_t)$. Then, based on the update in line 9 of Algorithm 2, we have:

$$\epsilon_t = (1 - \alpha_t)\epsilon_{t-1} + \alpha_t U_t + (1 - \alpha_t)W_t, \tag{19}$$

where $U_t = (g(\tau_t; \theta_t) - \nabla J(\theta_t))$ and $W_t = B(\tau_t^b; \theta_t^b)(\theta_t - \theta_{t-1}) - (\nabla J(\theta_t) - \nabla J(\theta_{t-1}))$. Let $\mathcal{H}_t$ be the history up to time $t$, i.e., $\mathcal{H}_t := \{\theta_0, \tau_0, \tau_1, b_1, \tau_1^b, \cdots, \tau_t, b_t, \tau_t^b\}$.

By taking the square $l_2$ norm of both sides of the above equation,

$$\begin{aligned}
\|\epsilon_t\|^2 &= (1 - \alpha_t)^2 \|\epsilon_{t-1}\|^2 + \alpha_t^2 \|U_t\|^2 + (1 - \alpha_t)^2 \|W_t\|^2 + \\
&\quad 2\alpha_t(1 - \alpha_t)\langle \epsilon_{t-1}, U_t\rangle + 2\alpha_t(1 - \alpha_t)\langle U_t, W_t\rangle + 2(1 - \alpha_t)^2\langle \epsilon_{t-1}, W_t\rangle \\
&\leq (1 - \alpha_t)^2 \|\epsilon_{t-1}\|^2 + 2\alpha_t^2 \|U_t\|^2 + 2(1 - \alpha_t)^2 \|W_t\|^2 + \\
&\quad 2\alpha_t(1 - \alpha_t)\langle \epsilon_{t-1}, U_t\rangle + 2(1 - \alpha_t)^2\langle \epsilon_{t-1}, W_t\rangle,
\end{aligned} \tag{20}$$

where we used Young's inequality in the first inequality for the following term: $2\alpha_t(1 - \alpha_t)\langle U_t, W_t\rangle \leq (1 - \alpha_t)^2 \|W_t\|^2 + \alpha_t^2 \|U_t\|^2$. Now, by taking expectations on both sides,

$$\begin{aligned}
\mathbb{E}[\|\epsilon_t\|^2] &\leq (1 - \alpha_t)^2 \mathbb{E}[\|\epsilon_{t-1}\|^2] + 2\alpha_t^2 \mathbb{E}[\|U_t\|^2] + 2(1 - \alpha_t)^2 \mathbb{E}[\|W_t\|^2] + \\
&\quad 2\alpha_t(1 - \alpha_t)\mathbb{E}[\langle \epsilon_{t-1}, U_t\rangle] + 2(1 - \alpha_t)^2 \mathbb{E}[\langle \epsilon_{t-1}, W_t\rangle] \\
&\overset{(a)}{\leq} (1 - \alpha_t)\mathbb{E}[\|\epsilon_{t-1}\|^2] + 2\alpha_t^2 \mathbb{E}[\|U_t\|^2] + 2\mathbb{E}[\|W_t\|^2] + \\
&\quad 2\alpha_t(1 - \alpha_t)\mathbb{E}[\langle \epsilon_{t-1}, U_t\rangle] + 2(1 - \alpha_t)^2 \mathbb{E}[\langle \epsilon_{t-1}, W_t\rangle] \\
&\overset{(b)}{\leq} (1 - \alpha_t)\mathbb{E}[\|\epsilon_{t-1}\|^2] + 2\alpha_t^2 \mathbb{E}[\|U_t\|^2] + 2\mathbb{E}[\|W_t\|^2],
\end{aligned} \tag{21}$$

(a) follows as $\alpha_t \leq 1$ for all $t \geq 0$.
(b) follows because the following two terms are zero. First, $\mathbb{E}[\langle \epsilon_{t-1}, U_t\rangle] = \mathbb{E}[\mathbb{E}[\langle \epsilon_{t-1}, U_t\rangle | \mathcal{H}_{t-1}]] = 0$ as $\epsilon_{t-1}$ is determined given $\mathcal{H}_{t-1}$ and $\mathbb{E}[U_t | \mathcal{H}_{t-1}] = 0$ since $g(\tau_t; \theta_t)$ is an unbiased estimation of $\nabla J(\theta_t)$. Second, $\mathbb{E}[\langle \epsilon_{t-1}, W_t\rangle] = \mathbb{E}[\mathbb{E}[\langle \epsilon_{t-1}, W_t\rangle | \mathcal{H}_{t-1}]] = 0$ because

$$\begin{aligned}
\mathbb{E}[W_t | \mathcal{H}_{t-1}] &= \mathbb{E}[B(\tau_t^b; \theta_t^b)(\theta_t - \theta_{t-1}) | \mathcal{H}_{t-1}] - (\nabla J(\theta_t) - \nabla J(\theta_{t-1})) \\
&= \mathbb{E}_{b_t}[\mathbb{E}_{\tau_t^b}[B(\tau_t^b; \theta_t^b)(\theta_t - \theta_{t-1}) | \mathcal{H}_{t-1}]] - (\nabla J(\theta_t) - \nabla J(\theta_{t-1})) \\
&\overset{(c)}{=} \mathbb{E}_{b_t}[\nabla^2 J(\theta_t^b)(\theta_t - \theta_{t-1}) | \mathcal{H}_{t-1}] - (\nabla J(\theta_t) - \nabla J(\theta_{t-1})) \\
&\overset{(d)}{=} \int_0^1 \nabla^2 J(b\theta_t + (1 - b)\theta_{t-1})(\theta_t - \theta_{t-1})db - (\nabla J(\theta_t) - \nabla J(\theta_{t-1})) = 0,
\end{aligned} \tag{22}$$

(c) It is due to the fact that for a given $\theta_t^b$, $B(\tau_t^b; \theta_t^b)$ is an unbiased estimation of $\nabla^2 J(\theta_t^b)$.
(d) The integral follows since $\theta_t^b = b_t \theta_t + (1 - b_t)\theta_{t-1}$ with $b_t$ uniformly distributed in $[0, 1]$, and its value is $\nabla J(\theta_t) - \nabla J(\theta_{t-1})$.

Now, from equation 21, we have:

$$\begin{aligned}
\mathbb{E}[\|\epsilon_{t-1}\|^2] &\overset{(a)}{\leq} \frac{1}{\alpha_t}\left(\mathbb{E}[\|\epsilon_{t-1}\|^2] - \mathbb{E}[\|\epsilon_t\|^2]\right) + \frac{2}{\alpha_t}\mathbb{E}[\|W_t\|^2] + 2\alpha_t \sigma_g^2 \\
&\overset{(b)}{\leq} \frac{1}{\alpha_t}\left(\mathbb{E}[\|\epsilon_{t-1}\|^2] - \mathbb{E}[\|\epsilon_t\|^2]\right) + 8\sigma_B^2 \frac{\eta_{t-1}^2}{\alpha_t} + 2\alpha_t \sigma_g^2,
\end{aligned} \tag{23}$$

(a) We use the bound in Lemma 1, i.e., $\mathbb{E}[\|U_t\|^2] \leq \sigma_g^2$.

(b) It has been shown that $\|B(\tau;\theta)\| \leq \sigma_B$ for any trajectory $\tau$ and $\theta \in \mathbb{R}^d$ (Shen et al., 2019). Therefore, $\|\nabla^2 J(\theta)\| = \|\mathbb{E}_\tau[B(\tau;\theta)]\| \leq \sigma_B$. Hence, $\nabla J(\theta)$ is Lipschitz with constant $\sigma_B$ and we have:

$$
\begin{aligned}
\|W_t\| &\leq \|B(\tau_t^b;\theta_t^b)(\theta_t - \theta_{t-1})\| + \|\nabla J(\theta_t) - \nabla J(\theta_{t-1})\| \\
&\leq \|B(\tau_t^b;\theta_t^b)\|\|\theta_t - \theta_{t-1}\| + \sigma_B\|\theta_t - \theta_{t-1}\| \\
&\leq 2\sigma_B\|\theta_t - \theta_{t-1}\|,
\end{aligned}
\tag{24}
$$

where the first inequality is due to the Lipschitzness of $\nabla J(\theta)$, and the second inequality results from the bound on $\|B(\tau_t^b;\theta_t^b)\|$.

Summing the both sides of equation 23 from $t=1$ to $t=T$, we have:

$$
\mathbb{E}\left[\sum_{t=1}^{T}\|\epsilon_{t-1}\|^2\right] \leq -\frac{\mathbb{E}[\|\epsilon_T\|^2]}{\alpha_T} + \frac{\mathbb{E}[\|\epsilon_0\|^2]}{\alpha_1} + \underbrace{\sum_{t=1}^{T-1}\left(\frac{1}{\alpha_{t+1}} - \frac{1}{\alpha_t}\right)\mathbb{E}[\|\epsilon_t\|^2]}_{(I)} + 8\sigma_B^2\underbrace{\sum_{t=1}^{T}\frac{\eta_{t-1}^2}{\alpha_t}}_{(II)} + 2\sigma_g^2\underbrace{\sum_{t=1}^{T}\alpha_t}_{(III)}
\tag{25}
$$

First note that $\mathbb{E}[\|\epsilon_0\|^2]/\alpha_1 \leq \sigma_g^2/\alpha_0$. Now, we bound the other terms in the right hand side of the above inequality:

(I): For the coefficient in the sum, we have: $1/\alpha_{t+1} - 1/\alpha_t = ((t+1)^{2/3} - t^{2/3})/\alpha_0 \leq 2t^{-1/3}/(3\alpha_0) \leq 2/(3\alpha_0)$ where we used the gradient inequality for the concave function $f(z) = z^{2/3}$. Therefore, this term can be bounded by: $(2/(3\alpha_0))\sum_{t=1}^{T-1}\mathbb{E}[\|\epsilon_t\|^2]$.

(II): $\sum_{t=1}^{T}\eta_{t-1}^2/\alpha_t = \eta_0^2/\alpha_0(1 + \sum_{t=1}^{T-1}(t+1)^{2/3}/t^{4/3}) \leq \eta_0^2/\alpha_0(1 + 2^{2/3}\sum_{t=1}^{T-1}t^{-2/3}) \leq 6\eta_0^2 T^{1/3}/\alpha_0$.

(III): $\sum_{t=1}^{T}\alpha_t = \alpha_0\sum_{t=1}^{T}t^{-2/3} \leq 3\alpha_0 T^{1/3}$.

Plugging these bounds in equation 25, we get:

$$
\mathbb{E}\left[\sum_{t=0}^{T-1}\|\epsilon_t\|^2\right] \leq CT^{1/3},
\tag{26}
$$

where $C := 3\alpha_0((48\sigma_B^2\eta_0^2 + \sigma_g^2)/\alpha_0 + 6\alpha_0\sigma_g^2)/(3\alpha_0 - 2)$.

The previous inequality yields that:

$$
\frac{1}{T}\sum_{t=0}^{T-1}\mathbb{E}[\|\epsilon_t\|] \leq \frac{1}{T}\sum_{t=0}^{T-1}\sqrt{\mathbb{E}[\|\epsilon_t\|^2]} \leq \sqrt{\frac{1}{T}\sum_{t=0}^{T-1}\mathbb{E}[\|\epsilon_t\|^2]} \leq \frac{\sqrt{C}}{T^{1/3}},
\tag{27}
$$

where we used Jensen's inequality in the second inequality above.

Now, if we average from $t = 0$ to $t = T - 1$ in equation 32 in Lemma 2 and then take expectations, we have:

$$
\begin{aligned}
\mathbb{E}\left[\frac{1}{T}\sum_{t=0}^{T-1}\|\nabla J(\theta_t)\|\right] &\leq \frac{8}{T}\sum_{t=0}^{T-1}\mathbb{E}[\|\epsilon_t\|] + \frac{3\sigma_B}{2T}\sum_{t=0}^{T-1}\eta_t + \frac{3}{T}\mathbb{E}\left[\sum_{t=0}^{T-1}\frac{J(\theta_{t+1}) - J(\theta_t)}{\eta_t}\right] \\
&\overset{(a)}{\leq} \frac{8\sqrt{C}}{T^{1/3}} + \frac{6\sigma_B\eta_0}{T^{2/3}} + \mathbb{E}\left[\frac{3}{T}\sum_{t=0}^{T-1}\frac{J(\theta_{t+1}) - J(\theta_t)}{\eta_t}\right] \\
&\leq \frac{8\sqrt{C}}{T^{1/3}} + \frac{6\sigma_B\eta_0}{T^{2/3}} + \mathbb{E}\left[\frac{3}{T}\left(\frac{J(\theta_T)}{\eta_{T-1}} - \frac{J(\theta_0)}{\eta_0} + \sum_{t=1}^{T-1}J(\theta_t)\left(\frac{1}{\eta_{t-1}} - \frac{1}{\eta_t}\right)\right)\right] \\
&\overset{(b)}{\leq} \frac{8\sqrt{C}}{T^{1/3}} + \frac{6\sigma_B\eta_0}{T^{2/3}} + \mathbb{E}\left[\frac{3}{T}\left(\frac{C_J(T-1)^{2/3}}{\eta_0} + \frac{C_J}{\eta_0} + \sum_{t=1}^{T-1}|J(\theta_t)|\left(\frac{1}{\eta_t} - \frac{1}{\eta_{t-1}}\right)\right)\right] \\
&\leq \frac{8\sqrt{C}}{T^{1/3}} + \frac{6\sigma_B\eta_0}{T^{2/3}} + \mathbb{E}\left[\frac{3}{T}\left(\frac{C_J(T-1)^{2/3}}{\eta_0} + \frac{C_J}{\eta_0} + \sum_{t=1}^{T-1}C_J\left(\frac{1}{\eta_t} - \frac{1}{\eta_{t-1}}\right)\right)\right] \\
&\overset{(c)}{\leq} \frac{8\sqrt{C}}{T^{1/3}} + \frac{6\sigma_B\eta_0}{T^{2/3}} + \mathbb{E}\left[\frac{3}{T}\left(\frac{C_J(T-1)^{2/3}}{\eta_0} + \frac{C_J}{\eta_0} + \frac{C_J(T-1)^{2/3}}{\eta_0}\right)\right] \\
&\leq \frac{8\sqrt{C}}{T^{1/3}} + \frac{6\sigma_B\eta_0}{T^{2/3}} + \frac{9C_J}{\eta_0 T^{1/3}},
\end{aligned}
\tag{28}
$$

$(a)$ We used the bound in equation 27 for the first term and $\sum_{t=0}^{T-1}\eta_t \leq 4\eta_0 T^{1/3}$.
$(b)$ $|J(\theta)| = |\mathbb{E}_{\tau\sim\pi_\theta}[R(\tau)]| = |\mathbb{E}_{\tau\sim\pi_\theta}[\sum_{h=0}^{H-1}\gamma^h r(s_h, a_h)]| \leq \mathbb{E}_{\tau\sim\pi_\theta}[\sum_{h=0}^{H-1}\gamma^h |r(s_h, a_h)|] \leq \mathbb{E}_{\tau\sim\pi_\theta}[R_0/(1-\gamma)] = R_0/(1-\gamma)$. Hence, we have: $|J(\theta)| \leq C_J$ for all $\theta \in \mathbb{R}^d$ where $C_J := R_0/(1-\gamma)$.
$(c)$ Due to $\sum_{t=1}^{T-1} 1/\eta_t - 1/\eta_{t-1} = 1/\eta_{T-1} - 1/\eta_0 \leq (T-1)^{2/3}/\eta_0$.

## B   Proof of Remark 3

According to equation 21, we have:

$$
\mathbb{E}[\|\epsilon_t\|^2] \leq (1-\alpha_t)\mathbb{E}[\|\epsilon_{t-1}\|^2] + 8\sigma_B^2\eta_{t-1}^2 + 2\alpha_t^2\sigma_g^2.
\tag{29}
$$

Let us define $Z_t := \mathbb{E}[\|\epsilon_t\|^2]$. Then, we can rewrite the above equation as follows:

$$
Z_t \leq (1-\alpha_t)Z_{t-1} + \frac{C_Z}{t^{4/3}},
\tag{30}
$$

where $C_z = 8 \times 2^{4/3}\eta_0^2\sigma_B^2 + 2\alpha_0^2\sigma_g^2$. Now, by induction, we will show that: $Z_t \leq C_Z'/t^{2/3}$ where $C_Z' = C_Z/(\alpha_0 - 2/3)$. It can be easily seen that for the base case $Z_1$, this statement holds. Now, for the induction step, suppose that $Z_{t-1} \leq C_Z'/(t-1)^{2/3}$ for some $t \geq 2$. Then, from above equation, we have:

$$
\begin{aligned}
\frac{C_Z}{t^{4/3}} &\overset{(a)}{\leq} \frac{\alpha_0 C_Z' - 2C_Z'/3}{t^{2/3}(t-1)^{2/3}} \\
&\implies \frac{2C_z'/3}{(t-1)t^{2/3}} \leq \frac{\alpha_0 C_Z'}{t^{2/3}(t-1)^{2/3}} - \frac{C_Z}{t^{4/3}} \\
&\overset{(b)}{\implies} C_Z'\left(\frac{1}{(t-1)^{2/3}} - \frac{1}{t^{2/3}}\right) \leq \frac{\alpha_0 C_Z'}{t^{2/3}(t-1)^{2/3}} - \frac{C_Z}{t^{4/3}} \\
&\implies \left(1 - \frac{\alpha_0}{t^{2/3}}\right)\frac{C_Z'}{(t-1)^{2/3}} + \frac{C_Z}{t^{4/3}} \leq \frac{C_Z'}{t^{2/3}} \\
&\overset{(c)}{\implies} (1-\alpha_t)Z_{t-1} + \frac{C_Z}{t^{4/3}} \leq \frac{C_Z'}{t^{2/3}} \\
&\overset{(d)}{\implies} Z_t \leq \frac{C_Z'}{t^{2/3}},
\end{aligned}
\tag{31}
$$

where $(a)$ is due to definition of $C'_Z$, $(b)$ is based on using gradient inequality for the concave function $f(z) = z^{2/3}$, i.e., $t^{2/3} - (t-1)^{2/3} \leq 2/3(t-1)^{-1/3}$, $(c)$ is according to induction hypothesis, and $(d)$ is due to equation 30.

## C  Supplemental Lemma

**Lemma 2** *Suppose that $\theta_t$'s are generated by executing Algorithm 2. Let $\epsilon_t := v_t - \nabla J(\theta_t)$. Then, at any time $t$, we have:*

$$\|\nabla J(\theta_t)\| \leq 8\|\epsilon_t\| + \frac{3\sigma_B \eta_t}{2} + \frac{3}{\eta_t}(J(\theta_{t+1}) - J(\theta_t)). \tag{32}$$

From $\sigma_B$-smoothness of $J(\theta_t)$ (Shen et al., 2019), we have:

$$\begin{aligned}
J(\theta_{t+1}) - J(\theta_t) &\geq \langle \nabla J(\theta_t), \theta_{t+1} - \theta_t \rangle - \frac{\sigma_B}{2}\|\theta_{t+1} - \theta_t\|^2 \\
&= \eta_t \left\langle \nabla J(\theta_t), \frac{v_t}{\|v_t\|} \right\rangle - \frac{\sigma_B}{2}\eta_t^2.
\end{aligned} \tag{33}$$

Regarding the first term above, we consider two cases, whether $\|\nabla J(\theta_t)\| \geq 2\|\epsilon_t\|$ or not. For the former case, we have:

$$\begin{aligned}
\left\langle \nabla J(\theta_t), \frac{v_t}{\|v_t\|} \right\rangle &= \frac{\|\nabla J(\theta_t)\|^2 + \langle \nabla J(\theta_t), \epsilon_t \rangle}{\|\nabla J(\theta_t) + \epsilon_t\|} \\
&\geq \frac{\|\nabla J(\theta_t)\|^2}{2\|\nabla J(\theta_t) + \epsilon_t\|} \\
&\geq \frac{\|\nabla J(\theta_t)\|^2}{2(\|\nabla J(\theta_t)\| + \|\epsilon_t\|)} \\
&\geq \frac{\|\nabla J(\theta_t)\|}{3} \\
&\geq \frac{\|\nabla J(\theta_t)\|}{3} - \frac{8}{3}\|\epsilon_t\|,
\end{aligned} \tag{34}$$

where in the first inequality, we used the bound $\langle \nabla J(\theta_t), \epsilon_t \rangle \geq -\|\nabla J(\theta_t)\|\|\epsilon_t\| \geq -\|\nabla J(\theta_t)\|^2/2$. For the latter case,

$$\begin{aligned}
\left\langle \nabla J(\theta_t), \frac{v_t}{\|v_t\|} \right\rangle &\geq -\|\nabla J(\theta_t)\| \\
&= \frac{\|\nabla J(\theta_t)\|}{3} - \frac{4\|\nabla J(\theta_t)\|}{3} \\
&\geq \frac{\|\nabla J(\theta_t)\|}{3} - \frac{8\|\epsilon_t\|}{3}.
\end{aligned} \tag{35}$$

Plugging the bound on $\langle \nabla J(\theta_t), v_t/\|v_t\| \rangle$ in equation 33, we get:

$$\begin{aligned}
J(\theta_{t+1}) - J(\theta_t) &\geq \eta_t \left( \frac{\|\nabla J(\theta_t)\|}{3} - \frac{8\|\epsilon_t\|}{3} \right) - \frac{\sigma_B}{2}\eta_t^2 \\
\implies \|\nabla J(\theta_t)\| &\leq 8\|\epsilon_t\| + \frac{3\sigma_B\eta_t}{2} + \frac{3}{\eta_t}(\nabla J(\theta_{t+1}) - \nabla J(\theta_t)).
\end{aligned} \tag{36}$$

## D  Comparison Using Another Variant of PR Metric

In this section, we report another variant of PR metric that takes into account the upper bounds of confidence intervals instead of the lower bounds. As can be seen in Table 3, SHARP still outperforms other algorithms in all environments considering the new variant of PR metric.

Table 3: Comparison of SHARP with other variance-reduced methods in terms of PR metric (The upper bounds of confidence intervals are considered in reporting PR metric).

|  | Reacher | Walker | Humanoid | Swimmer |
|---|---|---|---|---|
| HAPG | -23.05 | 127.94 | 230.48 | 88.31 |
| IS-MBPG | -20.78 | 296.85 | 309.51 | 30.32 |
| PAGE-PG | -17.25 | 259.36 | 402.20 | 25.05 |
| REINFORCE | -28.65 | 136.37 | 222.97 | 28.40 |
| VR-BGPO | -15.92 | 346.41 | 406.22 | 39.77 |
| SHARP (our algorithm) | **-10.23** | **440.70** | **441.27** | **172.42** |

## E    Details of Experiments

We used the default implementation of linear feature baseline from Garage library. The employed linear feature baseline is a linear regression model that takes observations for each trajectory and extracts new features such as different powers of their lengths from the observations. These extracted features are concatenated to the observations and used to fit the parameters of the regression with least square loss function. In the experiments, we used linear baseline for all the environments and methods.

In our reporting, at each iteration $t$, we generated trajectories according to the current policy until we collected $10k$ system probes. Then, we computed the average return based on the collected trajectories and considered this value for $10k \times t$ system probes.

In the following table, we provide the fine-tuned parameters for each algorithm. Batch sizes are considered the same for all algorithms. The discount factor is also set to 0.99 for all the runs.

|  | Reacher | Walker | Humanoid | Swimmer |
|---|---|---|---|---|
| Max horizon | 50 | 500 | 500 | 500 |
| Neural network sizes | $64 \times 64$ | $64 \times 64$ | $64 \times 64$ | $64 \times 64$ |
| Activation functions | Tanh | Tanh | Tanh | Tanh |
| IS-MBPG $lr$ | 0.9 | 0.3 | 0.75 | 0.3 |
| IS-MBPG $c$ | 100 | 12 | 5 | 12 |
| IS-MBPG $w$ | 200 | 20 | 2 | 25 |
| REINFORCE step-size | 0.01 | 0.01 | 0.001 | 0.01 |
| SHARP $\alpha_0$ | 1.5 | 5 | 5 | 4.5 |
| SHARP $\eta_0$ | 0.1 | 1 | 0.6 | 0.15 |
| PAGE-PG $p_t$ | 0.4 | 0.4 | 0.6 | 0.6 |
| PAGE-PG step-size | 0.01 | 0.001 | 0.0005 | 0.001 |
| HAPG step-size | 0.01 | 0.01 | 0.01 | 0.01 |
| HAPG $Q$ | 5 | 10 | 10 | 10 |
| VR-BGPO $lr$ | 0.8 | 0.75 | 0.8 | 0.75 |
| VR-BGPO $c$ | 25 | 25 | 25 | 25 |
| VR-BGPO $w$ | 1 | 1 | 1 | 1 |
| VR-BGPO $lam$ | 0.0005 | 0.0005 | 0.0025 | 0.0005 |

Table 4: Selected hyper-parameters for different methods.

In the following table, we provide the sets of considered values for hyperparameter tuning for each algorithm.

| | Set |
|---|---|
| IS-MBPG $lr$ | {0.1, 0.2, 0.3, 0.4, 0.55, 0.65, 0.75, 0.85, 0.9} |
| IS-MBPG $c$ | {5, 12, 20, 25, 50, 80, 100} |
| IS-MBPG $w$ | {2, 5, 10, 15, 25, 50, 100, 200} |
| REINFORCE step-size | {0.0005, 0.001, 0.01, 0.05, 0.1} |
| SHARP $\alpha_0$ | {1, 1.5, 2, 3, 3.5, 4.5, 4, 5} |
| SHARP $\eta_0$ | {0.1, 0.15, 0.2, 0.3, 0.4, 0.5, 0.6, 0.7, 0.8, 0.9, 1} |
| PAGE-PG $p_t$ | {0.2, 0.4, 0.5, 0.6, 0.7, 0.9} |
| PAGE-PG step-size | {0.0005, 0.001, 0.01, 0.05, 0.1} |
| HAPG step-size | {0.0005, 0.001, 0.01, 0.05, 0.1} |
| HAPG $Q$ | {5, 7, 10, 15, 20} |
| VR-BGPO $lr$ | {0.6, 0.7, 0.75, 0.8, 0.85 0.9, 0.95} |
| VR-BGPO $c$ | {15, 25, 35, 45, 50} |
| VR-BGPO $w$ | {1, 3, 5, 7, 10} |
| VR-BGPO $lam$ | {0.0005, 0.001, 0.0015, 0.002, 0.0025} |

Table 5: Sets of hyper-parameters for different methods.

