# OpenReview forum: "Momentum-Based Policy Gradient with Second-Order Information"
_TMLR — Accepted by TMLR_

### Review · Reviewer_BZdM · 2023-12-16

**Summary Of Contributions:**

# Summary

This paper presents a new algorithm called _Stochastic Hessian Aided Recursive Policy gradient_ (SHARP). The SHARP algorithm incorporates second-order, Hessian information into stochastic gradient descent (SGD) with momentum in order to reduce the variance of the policy gradient estimator. SHARP is able to reach first order stationary points (FOSP) without importance sampling and with relatively low sample complexity.  Unlike some other popular variance reduction methods for policy gradient algorithms, SHARP does not require large batch sizes or gradient checkpointing.

**Audience:**

Yes

**Broader Impact Concerns:**

No ethical concerns.

**Claims And Evidence:**

No

**Requested Changes:**

# Requested Changes

In order to secure my recommendation for acceptance, all the issues I listed in my _Main Argument_ would have to be addressed. This includes (but is not limited to):
1. Using confidence intervals for which all assumptions are satisfied.
2. The confidence intervals in (1) should not overlap and be high confidence estimates (>= 90% confidence) with many random seeds.
3. Reporting full confidence intervals, rather than only the average lower bound, while maintaining similar conclusions

To make the paper better, the issues in the _Small Things_ section above could be addressed. This is not critical for securing my recommendation for acceptance.

**Strengths And Weaknesses:**

# Main Argument

Recommendation: weak reject

I am concerned that there is not a sufficient empirical justification for the SHARP algorithm. A significant amount of details about the empirical study are missing -- details that are needed for a more thorough, complete evaluation. Below I will comment on my major concerns.

First, the paper does not outline the hyperparameter ranges that were tuned over for any algorithm. This information is vital in order for a successful evaluation of the empirical study. Were approximately the same number of hyperparameter settings tuned over for all algorithms? If some algorithms had more hyperparameter settings tuned over, they likely had an advantage over the other algorithms, as there were more avenues by which performance could be improved for such algorithms.

Second, I am unsure of the performance robustness (PR) metric that is reported for performance. This PR metric is basically the average lower bound of a 90% confidence interval over average return for each timestep in an agent-environment interaction. Because this metric only considers a lower bound, it hides a significant amount information, especially information about whether confidence intervals are overlapping. Here is equation 18 from the paper:

$$
PR_{A}^{\text{min}}(n) = \frac{1}{T} \sum_{t=1}^{T} LCI_{A}(n, t)
$$

We can also consider another metric:

$$
PR_{A}^{\text{max}}(n) = \frac{1}{T} \sum_{t=1}^{T} UCI_{A}(n, t)
$$

with $UCI_{A}(n, t)$ the upper bound of a 90% confidence interval over average return for $n$ runs of algorithm $A$. Why was this mteric not also reported? It would help to better gauge algorithm performance, especially when comparing algorithms.

It could be the case that for two algorithms $A$ and $B$ that $PR_{A}^\text{min}(n) > PR_{B}^\text{min}(n)$ but that $PR_{A}^\text{max}(n) < PR_{B}^\text{max}(n)$. In such a case, algorithm $B$ has a lower estimate of "worst-case" performance but a higher estimate of "best-case" performance, meaning that algorithm $B$ could really outperform $A$ on the problem. We don't have sufficient information to determine this. The $PR_A^\text{min}$ metric hides this information (and more). Further, I am unsure that comparing lower bounds of confidence intervals can sufficiently show that one algorithm is necessarily better than another. Why not provide the average lower and upper bounds for this confidence interval, as I have outlined above?

Furthermore, considering table 2, the $PR_{SHARP}^\text{min}$ value is not much higher than that of many of the other algorithms (especially on Humanoid and Walker), which seems to indicate that the confidence intervals which were used to generate these numbers do indeed overlap.

To compute the $PR_A^\text{min}$ metric, a confidence interval over average return is constructed over $n$ runs of the algorithm for each system probe. A system probe is simply a (state, action) pair. How was the average return computed for this (state, action) pair, since only a reward is available at this time? Is the episodic return used for the episode corresponding to the system probe under consideration?

Third, even if I was confident about the aforementioned metric, the paper is not clear about how confidence intervals are constructed. The paper mentions that 90% confidence intervals are used to compute the values in table 2, but there is no indication about what kind of confidence interval is used. I assume that t-distributed confidence intervals were used since this is what is common (please correct me if I am wrong). In this case, I am not confident that the distributional assumptions of the confidence interval are satisfied, and the reported confidence intervals may therefore be overly optimistic. Likely, 10 runs is not enough to satisfy the normality assumptions of such confidence intervals. This would also exacerbate the issue with table 2 that I mentioned above. Would it be possible to attain a significant number of more experimental runs (e.g. >=20 total)? Further, a likely better estimate of confidence could be obtained with confidence intervals which make no distributional assumptions about the data [1]. Does Figure 1 also use the same confidence intervals as reported in table 2?

Next, why were the default values not used for the maximum episode horizon for Walker, Swimmer, and Humanoid? The default values for each of these environments are significantly higher than what was used in the empirical study (default is 1000 timesteps). This design choice makes the results reported in the paper hard to compare with existing results in the literature which tend to use the default episode cutoffs. Further, reducing episode cutoffs can make these problems easier.

Finally, a number of things are not clear in the paper. First, $B(\tau_t^b, \theta_t^b)$ is used before it is clearly defined. Second, in _Remark 1_, why is it important to have an unbiased estimate of $\nabla J(\theta_t) - \nabla J(\theta_{t-1})$? Where does this difference in gradients come from? Is it related to equation 10? I am assuming this difference in gradients is from some theoretical result in Tran & Cutkosky's (2022) paper [2], but it is not clear why an unbiased estimate of this value is important. Equation 8 in Algorithm 1 is not clear. Are both gradient terms included in the sum?

# Small Things

Here are a list of small things which **did not affect the scoring of the paper** but can help to make the paper stronger:
- There are multiple grammatical errors in the paper. For example:
     - "In RL context" -> "In the RL context"
     - "In RL setting" -> "In the RL setting"
     - "In (Zhang et al., 2020)" - > "Zhang et al. (2020) state that..."
     - "distribution of starting state" -> "distribution of starting states"
     - "IS-MBPG... which is based on STROM" -> I think this should be "STORM".
     - "metric proposed in (Khorasani et al., 2023)" -> "Khorasani et al. (2023) propose the metric..."
- In many different places, vague terms are used. For example, at the beginning of the second paragraph in the introduction, it is stated that "Policy gradient methods are often used for obtaining good policies". What is a "good policy"?
- I found the first paragraph of the introduction hard to follow. There were multiple undefined terms in this paragraph as well (or terms were used before they were defined). For example, what are the "best actions"?
- I believe this is a typo, but in the last sentence of the paper it is claimed that SHARP attains remarkable performance on HalfCheetah. Yet, these results were never shown.

# References

[Cédric Colas, Olivier Sigaud, Pierre-Yves Oudeyer. How Many Random Seeds? Statistical Power Analysis in Deep Reinforcement Learning Experiments. 2018.](https://arxiv.org/abs/1806.08295)

[Hoang Tran, Ashok Cutkosky. Better SGD using Second-order Momentum. 2022.](https://arxiv.org/abs/2103.03265)

---

> ### Author Response · Authors · 2024-02-18
> **Review response**
>
> **Regarding tuning hyperparameters:**
>
> In the submitted manuscript, we explicitly mentioned the set of hyperparameters for each algorithm listed in Table 3 (Appendix E) and the values of tuned hyperparameters. In particular, SHARP (our algorithm) has only two hyper-parameters (the initial learning rate $\eta_0$ and the initial momentum weight $\alpha_0$), and its performance is fairly insensitive to $\alpha_0$. In fact, designing optimization algorithms with a few hyperparameters is more desirable in training learning models. It is especially vital in RL as performing grid search for several configurations of hyperparameters might be too time-consuming. In the context of stochastic optimization, there is a line of research on the so-called parameter-free optimization where the goal is to remove the need for hyperparameters with almost no knowledge of the problem properties (see the following references). As we show in Theorem 1, SHARP is parameter-free in the sense that $\alpha_0$ and $\eta_0$  do not depend on other parameters of the problem. Regarding the hyperparameter ranges, we added them to the Appendix E (Table 5).
>
> [1] Luo, Haipeng, and Robert E. Schapire. "Achieving all with no parameters: Adanormalhedge." In Conference on Learning Theory, pp. 1286-1304. PMLR, 2015.
>
> [2] Orabona, Francesco, and Tatiana Tommasi. "Training deep networks without learning rates through coin betting." Advances in Neural Information Processing Systems 30 (2017).
>
> [3] Levy, Kfir, Ali Kavis, and Volkan Cevher. "Storm+: Fully adaptive sgd with recursive momentum for nonconvex optimization." Advances in Neural Information Processing Systems 34 (2021): 20571-20582.
>
> **Regarding PR metric:**
>
> There are several options for selecting the best configuration of hyperparameters. For instance, the best hyperparameters can be selected according to the average return at the end of the horizon, or the maximum value of the average return achieved over the training phase, or the PR metric, considered in this paper. The current work is one of the few among the PG methods that explicitly mention how the hyperparameters are selected. Regarding PR metric, in (Khorasani et al., 2023), it has been observed that for some of the baselines (such as IS-MBPG), and for some specific configuration of hyperparameters, the average return quickly increases in the beginning but degrades drastically by the end of the horizon. To avoid picking such a configuration, PR metric is averaging over the training phase. Moreover, in order to be conservative when we report performance results, the lower bounds of confidence intervals are considered. As suggested by the reviewer, one can also report the performance based on the upper bounds. However, as  seen in Figure 1, the confidence interval for SHARP is small compared to the most baselines. Moreover, SHARP is the best algorithm in all environments, even if we report based on the upper bounds rather than the lower bounds. In Appendix D (Table 3), we also reported $PR^{max}$ for the considered algorithms.
>
> **Overlap in confidence intervals:**
>
> The learning curves corresponding to PR measures in Table 2, appear in Figure 1. Note that, in Humanoid, there is no overlap between the learning curve of SHARP and the second-best algorithm (VR-BGPO) after about 2 million system probes. This is also the case for Walker after about 7.5 million system probes. These results show that SHARP indeed outperforms the other algorithms in all the environments.
>
> **Regarding average return:**
>
> In each iteration $t$, we generated trajectories according to the current policy until we collected $10^4$ system probes. Then, we computed the average return based on the collected trajectories and considered this value for $10^4\times t$ system probes. Thus, the average return is reported only at the multiplication of $10^4$ system probes. In the revised version, we added this explanation in the Appendix E.
>
> **Regarding confidence intervals:**
>
> We utilized t-distribution to compute confidence intervals. The PR measures in Table 2 were based on the learning curves presented in Figure 1. As suggested by the reviewer, it would be preferable to use bootstrapped techniques which require about at least 30-40 runs, however, this process may take almost two months with our current computational resources. Nonetheless, Figure 1 clearly indicates that in all environments, the confidence intervals of SHARP and the baseline algorithms had almost no overlap. This indicates that 10 runs were sufficient to demonstrate that SHARP outperforms the baseline algorithms in all the considered environments.
>
> **Regarding horizon lengths:**
>
> In order to have a fair comparison with the previous work, we considered the same setup mentioned in the literature of variance reduction in PG methods.

---

> > ### Author Response · Authors · 2024-02-18
> > **Review response (second part)**
> >
> > **$B(\tau^b_t,\theta^b_t)$ is used before it is clearly defined.**
> >
> > We define the notation $B(\tau;\theta)$ exactly after eq. (5). Moreover, $\theta_t^b$ and $\tau_b^t$ are defined in the first paragraph of Section 3 just before using the notation $B(\tau^b_t,\theta^b_t)$. In the revised version, we remind the reader about $B(\tau;\theta)$ in Section 3 per your recommendation.
> >
> > **Regarding unbiased estimate of unbiased estimate of $\nabla J(\theta_t)-\nabla J(\theta_{t-1})$**
> >
> > In the context of stochastic optimization, a correction term in the form of $\nabla f(\theta_t,z)- \nabla f(\theta_{t-1},z)$ is used to produce the desirable decay rate in the variance of error term $\mathbb{E}[||\epsilon_t||^2]$. This allows the algorithm to achieve the sample complexity of $O(\epsilon^{-3})$. This approach has been considered in many variance reduction methods in the context of optimization. The approach in (Tran & Cutkosky 2022) is essentially the same except for the use of the Hessian vector product (HVP) term in the form of $\nabla^2 f(\theta_t,z_t)(\theta_t-\theta_{t-1})$. Please note that this term is **not** an unbiased estimate of $\nabla f(\theta_t) - \nabla f(\theta_{t-1})$ as the HVP is evaluated at point $\theta_t$. Thus, (Tran & Cutkosky 2022) provided a more complicated proof in order to obtain the sample complexity of $O(\epsilon^{-3})$. In SHARP, we evaluate HVP at a random point along the path between $\theta_{t-1}$ and $\theta_t$. As shown in Appendix A (see eq. (22)), this is indeed an unbiased estimate of $\nabla f(\theta_t) - \nabla f(\theta_{t-1})$, which enables us to provide a concise proof. Together with the well-designed deterministic time-varying learning rates and momentum weights, we show that SHARP is the first parameter-free and batch-free PG method that achieves $\epsilon$-FOSP with the sample complexity of $O(\epsilon^{-3}$).
> > In the revised version, we added more explanation in Remark 3 to show how the recursive inequality for the variance is derived and where the unbiasedness of $B(\tau^b_t,\theta^b_t)$ appears in the analysis.
> >
> > **Equation 8 in Algorithm 1 is not clear. Are both gradient terms included in the sum?**
> >
> > Yes, in the revised version, we added a parenthesis to remove any ambiguity.
> >
> >
> > **Regarding requested changes**
> >
> > We reported $PR^{max}$ in the revised version. Figure 1 shows that in all environments, the confidence intervals of SHARP and the baseline algorithms had almost no overlap. This indicates that 10 runs were sufficient to show that SHARP outperforms the baseline algorithms, even if we report based on the upper bounds rather than the lower bounds. Besides empirical performance, SHARP is the first parameter-free and batch-free PG method that achieves $\epsilon$-FOSP with the sample complexity of $O(\epsilon^{-3}$) without using IS weights. We kindly request the reviewer to reconsider their evaluation.

---

> ### Comment · Reviewer_BZdM · 2024-03-02
> **Response**
>
> I thank the authors for the clarifications made.
>
> My remaining concern with the paper is that the empirical evidence does not sufficiently support the claims made in the paper. With regards to the empirical results presented, I do not think that the confidence intervals reported provide an accurate estimate of confidence for a number of reasons.
>
> It is unlikely that the data satisfies the assumptions of the t-distributed confidence intervals, raising into question the validity of the reported confidence intervals. If the data does not satisfy such assumptions, then different confidence intervals -- for which assumptions are satisfied -- should be used. Further, the paper does not provide any evidence that the data does indeed satisfy the assumptions of the t-distributed confidence interval.
>
> Although using 10 runs seems to have been sufficient to produce non-overlapping confidence intervals, this does not necessarily indicate that 10 runs were sufficient to demonstrate performance differences between algorithms. The main issue is what I've previously outlined, whether the confidence interval assumptions are satisfied (e.g. see [1]). One could make an appeal to the Central Limit Theorem, but it is unclear whether 10 or 30 or 50 or even 100 runs are sufficient for the Central Limit Theorem to hold, and the only way to validate whether it does hold or not would be to obtain a (potentially infeasibly) significant number of more runs. Further, it is well-known that performance distributions in RL are highly stochastic, and this can lead to misleading confidence intervals if a sufficient number of experimental repetitions are not used [2].
>
> A further indication that the 10 experimental repetitions used in the paper are insufficient is the fact that the mean line exhibits more variation than the confidence intervals themselves in Figure 1. This indicates that more runs are needed for a confident estimate of mean performance. See [1] for further justifications.
>
>
> # References
>
>  [1] Empirical Design in Reinforcement Learning. Andrew Patterson, Samuel Neumann, Martha White, Adam White. 2023.
>
>  [2] Deep Reinforcement Learning that Matters. Peter Henderson, Riashat Islam, Philip Bachman, Joelle Pineau, Doina Precup, David Meger. AAAI. 2018.

---

> > ### Author Response · Authors · 2024-03-04
> >
> > To address the reviewer's concern, we considered the recommendation in [1] for reporting confidence intervals when few runs are available. In particular, it has been observed that in almost all Atari games, using percentile bootstrap, the estimated confidence interval contains the true mean with a probability of about 0.9 even with 10 runs. We updated Figure 1, Table 2, and Table 3 in the revision using percentile bootstrap. As can be seen, SHARP still outperforms all other considered algorithms in $PR$ (Table 2) and $PR^{max}$ (Table 3).
> >
> > [1] Agarwal, Rishabh, Max Schwarzer, Pablo Samuel Castro, Aaron C. Courville, and Marc Bellemare. "Deep reinforcement learning at the edge of the statistical precipice." Advances in neural information processing systems 34 (2021): 29304-29320.

---

> ### Comment · Reviewer_BZdM · 2024-03-05
> **Response**
>
> Thank you for the response and for considering using confidence estimates for which distributional assumptions are satisfied. Nevertheless, I am still not convinced that the reported levels of confidence are accurate.
>
> The referenced paper [1] considers confidence intervals which are constructed for scores aggregated over 26 games. Hence, when using 10 runs on a per-game basis, the total number of samples actually attained and used in the construction of the confidence interval in question would be 10 * 26 = 260. The idea here is that when one has only 10 runs on one game, but many games, then aggregating across games produces many samples from the joint distribution between performance and games. Because of this fact, the referenced paper notes that bootstrap CIs for scores on a per-game basis require many more runs than scores aggregated across games (i.e. the 10 runs discussed previously).
>
> Further, it is well-known that bootstrapping is sensitive and can produce inaccurate measures of confidence when the dataset size is small since bootstrapping assumes that re-sampling a dataset approximates sampling from the underlying distribution. This is also discussed in [1, 2], and is one reason why [1] mentions that many more runs are needed when constructing bootstrap CIs on a per-game basis compared to when aggregating across games.
>
> To be clear, I am not suggesting that hundreds of runs need to be completed on each environment for meaningful, statistically significant results. But, care needs to be taken to ensure reported CIs are valid and that assumptions are satisfied.
>
> # References
>
> [1] Agarwal, Rishabh, Max Schwarzer, Pablo Samuel Castro, Aaron C. Courville, and Marc Bellemare. "Deep reinforcement learning at the edge of the statistical precipice." Advances in neural information processing systems 34 (2021): 29304-29320.
>
> [2] How Many Random Seeds? Statistical Power Analysis in Deep Reinforcement Learning Experiments. Cédric Colas, Olivier Sigaud, Pierre-Yves Oudeyer. 2018.

---

> > ### Author Response · Authors · 2024-03-05
> >
> > [1] also evaluated 95% CI per game. As can be seen in Figure A.18, using percentile bootstrap, in almost all games, the fraction of CIs that contains the true mean score is about 0.9 even with 10 runs. We believe that this number of runs is sufficient to demonstrate the improvement of SHARP over previous methods in experiments, in addition to its theoretical guarantees.

---

> ### Comment · Reviewer_BZdM · 2024-03-05
> **Reviewer Response**
>
> In reference to Figure A.18, [1] recommends that these CIs require **at least** 20-30 runs **per-game** as opposed to 5-10 runs when reporting aggregated statistics over games. Further, Figure A.18 in [1] indicates that the construction of 95% confidence intervals with 10 runs will contain the true mean with 90% probability, but I expect this to be lower for 90% confidence intervals.

---

> > ### Author Response · Authors · 2024-03-05
> >
> > We would like to address the points raised with the following clarifications:
> > * The recommendation of 20-30 runs per game in [1] is for **95%** CI. To contain the true mean in computed CI with a probability of **0.9**, in almost all Atari games, ten runs were enough.
> > * The SHARP algorithm shows clear improvement in almost all environments from fairly the beginning of training. Although performing more runs makes CI more accurate, we believe that it does not change the conclusion.
> > * As we mentioned before, with our current computational resource, it takes about two months to have 20-30 more runs for each considered algorithm. Nevertheless, our code is publicly available, and our results can be easily reproduced or checked with more runs.
> > * Our experiments already involved more runs than recently published papers on policy gradient methods in TMLR having experiments in control tasks:
> >     * https://openreview.net/pdf?id=WFI9xhJrxF (5 runs)
> >     * https://openreview.net/pdf?id=jkTqJJOGMS (5 runs)
> >
> > We should emphasize that the primary contribution of the paper is to propose a batch-free and parameter-free PG algorithm achieving $\epsilon$-FOSP with the sample complexity of $O(\epsilon^{-3})$ without using IS weights. Therefore, SHARP does not require a strong assumption on IS weights, which is common in the literature.

---

### Review · Reviewer_wm5y · 2024-01-11

**Summary Of Contributions:**

This paper studies the problem of policy gradient optimization in reinforcement learning. The propose a new algorithm, with the primary purpose of delivering theoretical guarantees, that achieves a fast convergence rate to a stationary point. This is done with a variance reduction method that uses second order information and without further assumptions than what is typical. It is also done without importance-weighting, which is a common practical limitation. The algorithm is also shown to outperform peer methods.

**Audience:**

Yes

**Claims And Evidence:**

Yes

**Requested Changes:**

See above.

**Strengths And Weaknesses:**

Overall I think this is a good paper. It makes a distinct and solid contribution to the literature on variance-reduced policy gradient methods. The theory appears to be good given the existing literature surrounding this topic, and it achieves competitive rates under very mild conditions. The experiments show convincingly that the method outperforms the baselines considered. I don't have an explicit weaknesses to bring forth, but I do have some questions/suggestions. Perhaps these can be incorporated to further strengthen the paper.

Questions:

- Which the experiment baselines are 'parameter-free' and which ones aren't? How do you choose the hyperparameters for the ones that are not? It maybe helpful to highlight this in that section.
- You compared to REINFORCE, but not more modern algorithms like PPO and TRPO. Why not? It seems like it would be informative to the reader to see the difference been PPO (a go-to PG method) and this, even though the theoretical guarantees may not be as strong.
- What are the computational trade-offs between this algorithm and the others discussed in the related work and used as baselines? The proposed method includes a second-order calculation, which can be prohibitive for larger models.
- I believe the page limit is 12, correct? I thought it might be helpful to include a sketch of the proof or at least highlight some of the core ideas that allow the analysis to through (beyond the algorithmic ones already discussed). What do you think?

---

> ### Author Response · Authors · 2024-02-18
> **Review response**
>
> Which the experiment baselines are 'parameter-free' and which ones aren't? How do you choose the hyperparameters for the ones that are not? It maybe helpful to highlight this in that section.
>
> * None of the baselines are parameter-free. As mentioned in the Experiment section, we considered a metric (called PR) to capture both the performance and robustness of an algorithm. For each configuration of hyperparameters, we ran each algorithm five times and computed the PR metric. We selected the configuration maximizing PR metric and reported the performance of the algorithm for the selected configuration based on 10 new runs. The selected hyperparameters for each algorithm are given in Appendix E.
>
> You compared to REINFORCE, but not more modern algorithms like PPO and TRPO. Why not? It seems like it would be informative to the reader to see the difference been PPO (a go-to PG method) and this, even though the theoretical guarantees may not be as strong.
>
> * The performance of TRPO is better than the algorithms considered in the experiments. Following the experimental setup of previous work, we limited our comparison to algorithms that have a theoretical guarantee of achieving $\epsilon$-FOSP with a sample complexity of $O(\epsilon^{-3})$. To the best of our knowledge, there is no theoretical guarantee for TRPO in general nonconvex setting.
>
> What are the computational trade-offs between this algorithm and the others discussed in the related work and used as baselines? The proposed method includes a second-order calculation, which can be prohibitive for larger models.
>
> * SHARP requires computing one Hessian vector product per iteration. As we mentioned at the end of Section 2, this can be done in $O(Hd)$ (similar to the computational complexity of obtaining the gradient vector) by executing Pearlmutter’s algorithm. Therefore, the computational cost of SHARP per iteration is in the same order as first-order methods such as REINFORCE. We emphasized this in the revised version in Section 3.
>
> I believe the page limit is 12, correct? I thought it might be helpful to include a sketch of the proof or at least highlight some of the core ideas that allow the analysis to through (beyond the algorithmic ones already discussed). What do you think?
>
> * In the revised version, we sketched the main steps of the proof in Remark 3, showing in particular how the recursive inequality for the variance is derived.

---

### Review · Reviewer_H6Lx · 2024-02-06

**Summary Of Contributions:**

The authors propose a new variance-reduced policy gradient method by use of the second-order information. This new method improves existing methods by removing the requirement of importance sampling and careful checkpoint. The authors prove sublinear rate to obtain an near-optimal stationary point. Experiments are also provided to show the effective performance of the proposed method.

**Audience:**

Yes

**Claims And Evidence:**

Yes

**Requested Changes:**

Overall, the result is nicely presented. Here are a few questions for consideration.

- In SHARP, is the choice paramters $(\eta_t,\alpha_t)$ in same order optimal?

- What's the policy parametrization used in experiments? How do you check the assumption made on the policy?

**Strengths And Weaknesses:**

Strengths:

(1) The proposed method is clearly explained. The authors also made a nice literature review that is helpful to read the advantages of the proposed method.

(2) The proposed method introduces a middle-point policy to get an unbiased gradient difference. Due to such second-order information, there is no need of checking points for sampling batches.

(3) The learning parameters do not reply on problem parameters, which makes the proposed method more practical.

Weaknesses:

(1) The sample complexity is sub-optimal.

(2) Some assumptions can be strong, such as Assumption 2, although it is often assumed in literature.

---

> ### Author Response · Authors · 2024-02-18
> **Review response**
>
> The sample complexity is sub-optimal.
>
> * As we mentioned in Remark 5, in the context of stochastic optimization, it is shown that under some assumptions on the objective function and stochastic gradients, the sample complexity of $O(\epsilon^{-3})$ is optimal in order to achieve $\epsilon$-FOSP and it cannot be improved with stochastic $p$-th order methods with $p\geq 2$. To the best of our knowledge, there is no lower bound in the context of reinforcement learning to converge to $\epsilon$-FOSP for general non-convex $J(\theta)$. Nevertheless, as can be seen in Table 1, the best achievable rate in our setting in the literature is $O(\epsilon^{-3})$ (achieved by SHARP) and it might very well be the optimal rate.
>
>
> Some assumptions can be strong, such as Assumption 2, although it is often assumed in literature.
>
> * Indeed as the reviewer points out these are common assumptions in the literature to facilitate the analysis of stochastic variance reduction techniques in RL. To give an example of class of policies that satisfy these assumptions, consider the Gaussian policy with a fixed standard deviation $\sigma$ as follows:
> \begin{equation*}
>     \pi_{\theta}(a|s)=\frac{1}{\sqrt{2\pi}\sigma}\exp\left(-\frac{(\theta^T\mu(s)-a)^2}{2\sigma^2}\right),
> \end{equation*}
> where $\mu(s):\mathcal{S}\rightarrow \mathbb{R}^d$ is a feature map. It can be easily seen that $\nabla \log
> \pi_{\theta}(a|s)=-(\theta^T\mu(s)-a)\mu(s)/\sigma^2$ and $\nabla^2\log\pi_{\theta}(a|s)=-\mu(s)^T\mu(s)/\sigma^2$. Therefore, Assumption 2 is satisfied for Gaussian policies if the action space, the domain of the parameter vector, and the feature map are bounded.
>
> Requested Changes:
>
> In SHARP, is the choice $(\alpha_t,\eta_t)$ parameters in same order optimal?
>
> * In order to achieve the sample complexity of $O(\epsilon^{-3})$, $\eta_t$ and $\alpha_t$ should decay with the same rate in $t$ in our analysis.
>
> What's the policy parametrization used in experiments? How do you check the assumption made on the policy?
>
> * As mentioned in the Experiment section, in all experiments, we considered Gaussian policies parameterized with neural networks having two layers of 64 neurons each. Let $\mu_{\theta}(s)$ be the output of the network representing the mean of action. In our experiments, we observed that $\mu_{\theta}(s), ||\nabla \mu_{\theta}(s)||$, and $||\nabla^2 \mu_{\theta}(s)||$ remained bounded. This ensures Assumption 2 is satisfied along the trajectory of optimization. Regarding Assumption 1, in the experiments, the reward function is bounded.

---

### Author Response · Authors · 2024-02-18
**Revised paper**

We thank all reviewers for their feedback. We addressed all the comments in the review responses.  We believe that addressing the reviewers' comments has resulted in a better presentation and readability of the paper. The updated parts in the revised paper are highlighted in blue for clarity.

---

### Decision · Action_Editor_MNRi · 2024-04-01

**Recommendation:** Accept as is

**Comment:**

Overall, the proposed algorithm SHARP is both theoretically sound and appears to perform well in experiments on some MuJoCo tasks. The reviewers are generally positive about the contributions of the paper. Therefore, I recommend acceptance.

I encourage the authors to further revise the paper based on the reviewers' suggestions in the camera ready version. For example, the authors may consider having more runs for computing the bootstrap confidence intervals, as raised by Reviewer BZdM.

**Audience:**

The results should be of interest to many areas such as deep RL, RL theory, and optimization.

**Claims And Evidence:**

This paper propose a new variance-reduced policy gradient algorithm for reinforcement learning. The algorithm SHARP uses second-order information in its update. Theoretically, the algorithm converges in $O(1/\epsilon^3)$ iterations, matching previous work while having some additional practical benefits such as no importance weighting and no need of policy checkpoints. Experiments on MuJoCo validates the fast convergence of the proposed algorithm. The advantages of the algorithm appear to be well supported by both theory and experiments.